# Adolescents with Normal Weight Obesity Have Less Dry Lean Mass Compared to Obese Counterparts

**DOI:** 10.3390/ijerph22020171

**Published:** 2025-01-27

**Authors:** Ann F. Brown, Ariel J. Aguiar Bonfim Cruz, Malayna G. Schwartz, Samantha J. Brooks, Alexa J. Chandler

**Affiliations:** 1Department of Movement Sciences, College of Education, Health & Human Sciences, University of Idaho, Moscow, ID 83843, USA; afbrown@uidaho.edu (A.F.B.); ariela@uidaho.edu (A.J.A.B.C.); sbrooks.phd@gmail.com (S.J.B.); 2WWAMI Medical Education Program, University of Idaho, Moscow, ID 83843, USA; mhambly@uw.edu

**Keywords:** normal weight obesity, adolescents, dietary intake

## Abstract

Normal weight obesity (NWO) is a condition characterized by a normal body mass index (BMI; 18.5–24.9 kg·m^−2^) yet excess body fat. Those with this condition have an increased risk of cardiometabolic diseases associated with obesity. The prevalence of NWO is not well investigated in adolescents, particularly in the United States. This study examined the prevalence of NWO and dietary behaviors among adolescents aged 14–19 years old (n = 139) who live in a rural area in the United States. Data were collected from December 2019 through February 2020. Body composition was assessed via bioelectrical impedance analysis and diet was assessed using an Automated Self-Administered 24 h food recall questionnaire. Participants were categorized by BMI and body fat percentage as NWO, normal weight lean (NWL), or obese (OB). The sample prevalence of NWO was 13.6%, with girls having a higher prevalence (22.2%) than boys (1.8%). Those with NWO had significantly lower dry lean mass than OB (*p* = 0.02), but there were no differences between NWL and OB (*p* = 0.08). There was significantly higher caloric intake (*p* = 0.02) among NWL compared to OB, and NWL consumed more fiber than both NWO (*p* = 0.02) and OB (*p* = 0.03). Overall, this study gives us a better understanding of the prevalence of NWO in the adolescent population and the dietary habits associated with each group. Those with NWO may be at increased risk for negative long-term health outcomes commonly associated with obesity. Additionally, the higher caloric intake among NWL was unexpected and should be investigated further.

## 1. Introduction

Childhood obesity is a public health crisis afflicting ~15 million American children and adolescents [1]. These children and adolescents are at increased risk of a myriad of health problems, including type 2 diabetes, hypertension, and cardiovascular diseases [2]. Obesity is defined as a Body Mass Index (BMI; kg·m^−2^) above the 95th percentile for adolescents. BMI is typically used to classify obesity because of its simplicity and low cost; however, this method fails to identify approximately 29% of individuals with bona fide obesity because relative amounts of body fat (BF) cannot be determined [3,4]. Due to this, there is a hidden population of individuals misclassified as ‘normal weight’ based on BMI, yet who possess excess adiposity, resulting in the ‘normal weight obese’ (NWO) phenotype [5]. Despite appearing similar to normal weight lean (NWL) adolescents, those with NWO have similar health risks as individuals with obesity and suffer from chronic inflammation as seen in obesity [4,6,7,8,9,10]. This misclassification may be particularly concerning among adolescents entering puberty when body composition drastically changes [11]. Current literature indicates prevalence of NWO among adolescents in South America and Europe ranges from 6.8 to 55.6% with no reportedly known prevalence in the United States [10,12,13,14,15,16,17]. Understanding the prevalence of adolescents with NWO is critical to prevent detrimental obesity-related health complications later in life.

Energy intake and expenditure are key contributors to the development of obesity in early childhood [18]. However, it is frequently observed that those with obesity have similar energy intake to those who are nonobese [18]. Energy intake and appetite control are hypothesized to be associated with deficits in lean mass (LM) which ultimately drives hyperphagia [19]. These deficits not only reduce energy expenditure but trigger increased energy intake, resulting in fat mass gain [19,20]. The rebound of fat mass from deficits in LM has been defined as collateral fattening and may be a contributing factor in the development of NWO [19]. Those with NWO subsequently have reduced relative LM and muscle strength [21]; however, it remains unclear if this phenotype is associated with energy intake, especially in an adolescent population. Therefore, clarity on the relationship between energy intake and BF and LM in adolescence is needed.

In addition to total energy consumption, dietary quality is an important aspect to consider when evaluating factors contributing to NWO [22]. Those with NWO appear to consume more processed foods and sugar-sweetened beverages compared to their NWL counterparts [22]. Low fruit and vegetable consumption coupled with high intake of refined, fried, and fatty foods can lead to nutritional deficiencies [23]; previous research found vitamin D deficiencies in those with NWO [24]. Further, daily recommended intakes are based on data from healthy individuals and do not account for the increased metabolic demands that occur with obesity, which may increase nutritional deficiency risk among those with NWO [25].

Identifying adolescents with NWO is important, as the development of obesity-related comorbidities at a young age may be particularly harmful and detrimental to health during adulthood [26]. It is imperative to better understand the true prevalence of childhood obesity by measuring BF, rather than BMI, due to the profound impact of excess adiposity on long-term health. Further, understanding differences in energy intake between NWO and NWL adolescents may provide insight into the dietary factors associated with NWO. Therefore, the purpose of this study was (1) to assess the prevalence of NWO among adolescents living in a rural area of the United States and (2) to evaluate and compare energy intake and dietary behavior between NWO, NWL, and obese (OB) adolescents. It was hypothesized that the prevalence of NWO would be high among the sample of adolescents given the known barriers to high-quality foods in rural areas. It was also hypothesized that those with OB would exhibit the greatest caloric consumption and those with NWO would consume less protein than NWL, as protein is essential for lean mass.

## 2. Materials and Methods

### 2.1. Participants

Adolescents (n = 139), ages 14–19 years, were recruited from six different high schools in a rural area of the United States. Researchers worked directly with school administrators to receive permission to contact families with a student enrolled in the participating schools. Families were recruited via email, and those who expressed interest in the study were given an informational packet explaining the study methods, risks, and benefits of participation prior to giving parental informed consent and child assent. The study was approved by the University of Idaho Institutional Review Board (IRB #: 19-119) and conformed to the recommendations of the Declaration of Helsinki.

### 2.2. Study Design

This cross-sectional study took place at the high schools from which participants were recruited. Data were collected from December 2019 through February 2020. Data collection was interrupted by the global pandemic of the 2019 novel coronavirus (COVID-19). Therefore, data were planned to be collected across 12 schools, but only 7 are included in this study. Participants completed a single testing session during the school day in a private room at their respective school. Prior to measurements, participants were asked to void their bladder, then consume 355 milliliters of water to ensure standard hydration. Height was measured using a travel stadiometer (BSM 170, InBody USA, Audubon, PA, USA) to the nearest centimeter. Body composition was assessed using bioelectrical impedance analysis (BIA) with a digital scale (InBody 270, InBody, USA, Audubon, PA, USA). Measures included body mass (kg), total body water (TBW; kg), dry lean mass (DLM; kg), fat mass (kg), and body fat (%). Participants’ hands and feet were wiped clean prior to stepping onto the BIA device, per manufacturer instructions. The BIA assessment lasted approximately two minutes, and the results screen was covered to prevent the participant from seeing their results. Lastly, participants completed a medical history questionnaire and dietary assessment (24 h recall) using the Automated Self-Administered Dietary Assessment Tool (ASA-24). Following the questionnaire completion, participants were offered a snack to thank them for their participation.

### 2.3. Data Analysis

Participants were classified by gender, BMI percentiles [27], and BF percentage as NWL, NWO, or OB (Figure 1). Participants were classified as NWL if their BMI was between the 5th and 85th percentile; BF percentage was ≤24.9% for boys and ≤27.9% for girls, while those classified as NWO fell within the same BMI percentiles but had BF percentages of ≥25% for boys and ≥28% for girls [16]. Participants were classified as OB if BMI was above the 85th percentile and BF was ≥25% for boys and ≥28% for girls [16]. Participants who did not fit any of these three classifications (n = 16) were excluded from analyses by weight classification.

Descriptive analyses were used to assess means and standard deviations by weight classification. Data were assessed for normality using Shapiro–Wilkes Test. A one-way analysis of variance (ANOVA) was used to assess the differences in body composition and dietary measures between weight classifications for the entire sample while accounting for sex and between-weight classifications for only girls and for only boys. Post hoc tests using Tukey’s correction factor were conducted when significant main effects were found, and estimates with standard error (SE) are displayed where appropriate. Spearman’s correlations were used to assess these relationships for boys and girls separately. An alpha level of 0.05 was used to determine statistical significance for all analyses. Statistical analyses were conducted in R (version 4.4.1) with lme4 version 1.1–35.5 [28] and emmeans version 1.10.4 [29], and figures were created with ggplot2 [30].

## 3. Results

### 3.1. Demographics and Body Composition

Participant group demographics are reported in Table 1, and differences in body composition variables are displayed in Figure 2, Figure 3 and Figure 4. While there were no differences in weight between those classified as NWO and NWL (*p* = 0.37), NWO adolescents had higher BMI (*p* = 0.006), BF percentage (*p* < 0.001), and fat mass (*p* < 0.001) values compared to NWL adolescents. Further, those classified as NWO had significantly lower DLM (*p* = 0.02) and TBW (*p* = 0.02) values than those classified as OB, but there were no differences between NWL and OB adolescents (DLM: *p* = 0.08; TBW: *p* = 0.09) or NWL and NWO (DLM: *p* = 0.4; TBW: *p* = 0.4).

### 3.2. Macronutrient Intake

The relationships between body composition and macronutrient intake are displayed in Table 2. Overall, macronutrient intake was negatively associated with body composition variables, suggesting that those with lower energy intake were leaner. Macronutrient intake showed stronger associations with BF compared to BMI in the overall sample. Additionally, there were stronger associations with BMI and BF when evaluating macronutrient intake relative to body mass rather than absolute intake for boys.

Overall, the NWL group consumed more total calories (*p* = 0.02), total CHO (*p* = 0.03), and total fat (*p* = 0.04), compared to the OB group (Table 3). NWL adolescents consumed more polyunsaturated fat than the OB group (+8.5 [3.0] g, *p* = 0.01), but there were no significant differences for monounsaturated fat, saturated fat, or dietary cholesterol. There were no significant differences in sugar consumption between weight classes (*p* = 0.25), but NWL adolescents consumed more fiber than both the NWO (+107.4 [39.6] g, *p* = 0.02) and OB (+85.6, [33.7] g, *p* = 0.03) groups. On a relative basis, NWL adolescents consumed more total calories, protein, CHO, and fat (Figure 5).

While there were no differences in absolute calorie or macronutrient intake for both boys and girls, there were differences within sex when assessed relative to body weight. Girls classified as NWL consumed more total kcal·kg^−1^ (+16.7 [4.5], *p* = 0.001), CHO (+1.9 [3.3] g·kg^−1^, *p* = 0.004), protein (+0.66 [3.0] g·kg^−1^, *p* = 0.01), and fat (+0.76 [0.2] g·kg^−1^, *p* = 0.003) than OB girls. Further, NWL girls consumed significantly more fiber than both NWO (+120.6 [41.6] g, *p* = 0.01) and OB girls (+120.3 [41.6] g, *p* = 0.01), but there were no differences in monounsaturated fat, polyunsaturated fat, saturated fat, cholesterol, or added sugars intake (*p* > 0.05). In contrast, boys classified as NWL only consumed more kcal·kg^−1^ (+18.1 [6.5], *p* = 0.02) and CHO (+2.4 [0.9], *p* = 0.02) than boys classified as OB.

### 3.3. Micronutrient Intake

The relationships between body composition and micronutrient intake are displayed in Table 4. There were minimal significant relationships for the whole sample, as only calcium intake was negatively correlated with BF. Similarly, only sodium was negatively correlated with BF in boys. In contrast, magnesium, zinc, and iron were all negatively correlated with BF in girls, suggesting that those consuming higher micronutrient quantities had less fat mass. Additionally, both magnesium and iron were inversely correlated with BMI in girls, but there were no significant relationships between BMI and micronutrients in boys.

For vitamins, NWL adolescents consumed significantly more riboflavin (+146.1 mg [58.8] mg, *p* = 0.04) and niacin (+198.7 [80.3] mg, *p* = 0.04) than NWO adolescents and significantly more niacin (+164.1 [68.3] mg, *p* = 0.047) and vitamin K (+63.9 [25.9] mcg, *p* = 0.04) than OB adolescents. For minerals, NWL adolescents consumed significantly more magnesium than the NWO group (+1052 [402] mg, *p* = 0.03) and the OB group (+854 [342 mg], *p* = 0.04), and significantly more iron than the NWO group (+851 [338] mg, *p* = 0.04). Further, girls consumed less calcium (−402 [150] mg, *p* = 0.008) and phosphorus (−368 [183], *p* = 0.047) than boys, but there were no sex differences for any other vitamin or mineral. Despite a few significant differences, mineral consumption was assessed relative the RDA to assess diet quality. Results indicated over half (total = 57% [girls = 54%; boys = 60%]) of participants met the RDA for zinc, but this was not the case for calcium, iron, magnesium, phosphorus, or potassium (Figure 6A–E). Further, 45% of participants (girls = 43%; boys = 47%) consumed excess sodium, with 43% of NWL adolescents, 38% of NWO adolescents, and 59% of OB adolescents consuming above the recommended amounts (>2300 mg).

## 4. Discussion

The current study found that 13.6% of the current sample was classified as NWO with a higher prevalence among girls than boys. These findings align with prior research reporting prevalence ranging from 6 to 55.6% in adolescents throughout South America and Europe [12,14,15,16,17]. Despite the feasibility of BMI, relying on this measure alone does not capture those classified as NWO and therefore does not allow this population to receive necessary medical attention and nutritional advice. For instance, 22% of girls in the present study were classified as NWO and would therefore be treated as healthy if only looking at BMI. However, those with NWO are at greater risk for comorbidities associated with obesity and proper education is imperative to reduce these risks. Further, this study found more associations between dietary habits and body fat than BMI, suggesting body fat, rather than BMI, is influenced by dietary habits and diet quality. This finding is important as concerns regarding diet in those with NWO may be overlooked, as they are classified as normal weight, but dietary education and interventions would likely improve future health-related outcomes in those with NWO.

Contrary to our hypothesis, those classified as NWL reported consuming the most total daily calories and subsequently higher amounts of each macro- and micronutrient, while those classified as OB reported the lowest total caloric intake. Despite significantly different caloric intake, lean body tissue was not significantly different between the NWL and OB groups, but was significantly lower in the NWO group compared to those classified as OB. This is particularly interesting given that the previous literature suggests deficits in fat-free mass, not excess fat mass, are associated with increased total energy intake and therefore obesity development [19,31,32,33]. Given average micronutrient intake was below the RDA in this study’s sample, improving fat-free mass, rather than focusing on reducing fat mass, may be a more feasible and sustainable recommendation which may lead to long-term health improvements in adolescents with both NWO and OB.

Despite reporting the lowest total calorie intake, most adolescents classified as OB reported excess sodium consumption. Excess sodium intake is a well-established risk factor for the development of cardiovascular diseases, and excess adipose tissue may further increase risk [34,35]. Further, a smaller percentage of NWO adolescents met the RDA for magnesium, zinc, and iron than OB adolescents. Micronutrient deficiencies may play an important role in the development of obesity in adolescents. While those who present with obesity may appear to be meeting the requirements for micronutrients, practitioners need to consider that the acceptable ranges were developed based on normal-weight individuals, and therefore do not account for the increased metabolic demands occurring with obesity [36]. However, the present sample shows even those classified as OB were deficient in micronutrients, likely driven by the overall low caloric intake. Micronutrient deficiency in adolescence is concerning, as this is a time of bone mineralization, rapid growth, and major physiological changes [37]. The low micronutrient intakes in this sample suggest poor diet quality, which may be related to household socioeconomic status or access to fresh fruits and vegetables, although information regarding these potential barriers was not collected.

## 5. Strengths and Limitations

Overall, the findings from this study contribute to our understanding of the prevalence and behavior characteristics of NWO in adolescents. The ability to assess body fat allowed the researchers to further explore the relationship between body composition and dietary habits that would not be captured by BMI alone, as evidenced by the correlational results. Data were collected from adolescents at 7 different high schools, although 12 schools originally expressed interest in the study. However, data collection was interrupted by the COVID-19 pandemic, leaving researchers with a smaller sample than originally intended.

Despite the strengths of this study’s methodology, the limitations cannot be overlooked. Data for the present study were only collected from adolescents in a single region of the United States. National census data indicate the participants in this study were from predominately white areas of the United States where only 30% of adults hold a bachelor’s degree or higher and median household incomes are below the national average [38]. Therefore, generalizability to adolescents across the entire United States is limited and more data are needed to understand how these demographic factors relate to NWO prevalence.

While measuring body composition and dietary intake is a strength as many studies rely solely on BMI, the measurements used in the present study have limitations. First, body composition measurements were assessed on a BIA device for ease of transportation. Body composition assessments have low-to-medium variability between devices, especially when categorizing NWO, and, thus, it is difficult to compare our prevalence results to other studies assessing NWO using a different assessment device. Second, dietary recall tools assume participants are truthful when reporting. Even if the participants in the present study were truthful, the dietary recall was only conducted on a single day and was likely not an accurate representation of dietary habits for some participants. Future research should consider administering a minimum of two dietary recall questionnaires to ensure reliability, although some recommend up to nine dietary recalls may be needed [39].

## 6. Conclusions

Overall, 13.6% of the current sample presented with NWO, with a higher prevalence among girls compared to boys. Those with NWO presented with comparable body fat to those with obesity, yet had significantly less lean mass, which may indicate negative long-term health outcomes. Further, the correlational analyses suggest that body fat is a more robust indicator of dietary habits than BMI, indicating BMI alone is not enough to determine health status in adolescents.

## Figures and Tables

**Figure 1 ijerph-22-00171-f001:**
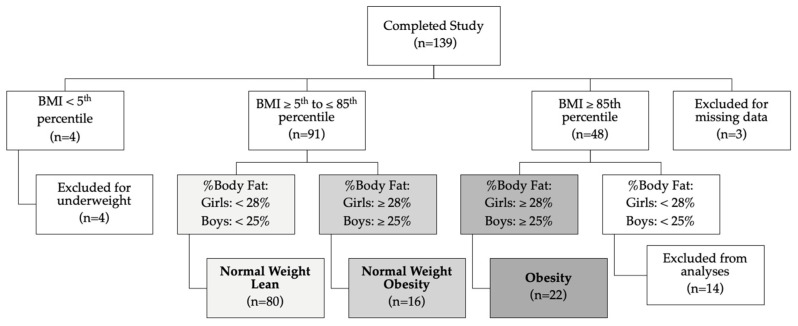
Schematic of group allocations. 84.9% of participants who completed the study were included in the data analysis (n = 118) based on body mass index.

**Figure 2 ijerph-22-00171-f002:**
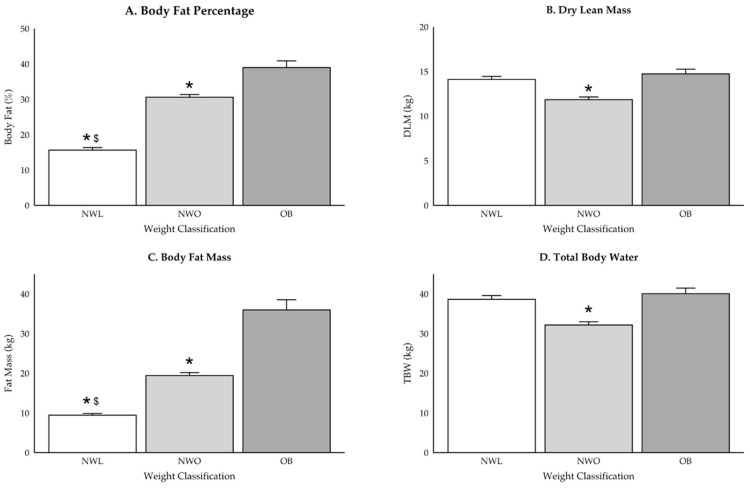
(**A**–**D**) Differences in body composition for the whole sample. * indicates significant difference from OB (*p* < 0.05) and ^$^ indicates significant difference from NWO (*p* < 0.05).

**Figure 3 ijerph-22-00171-f003:**
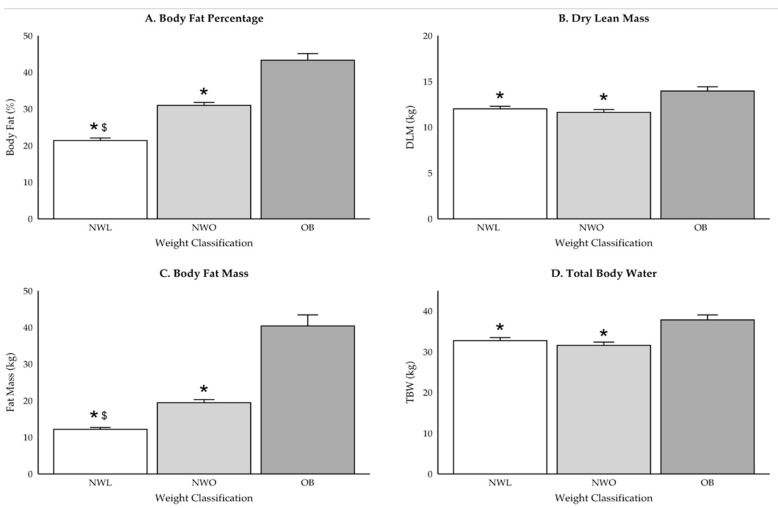
(**A**–**D**) Differences in body composition for girls only. * indicates significant difference from OB (*p* < 0.05) and ^$^ indicates significant difference from NWO (*p* < 0.05).

**Figure 4 ijerph-22-00171-f004:**
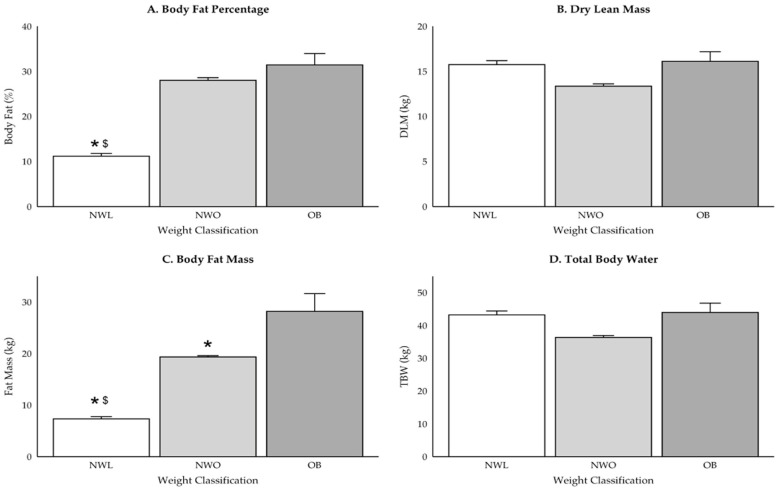
(**A**–**D**) Differences in body composition for boys only. * indicates significant difference from OB (*p* < 0.05) and ^$^ indicates significant difference from NWO (*p* < 0.05).

**Figure 5 ijerph-22-00171-f005:**
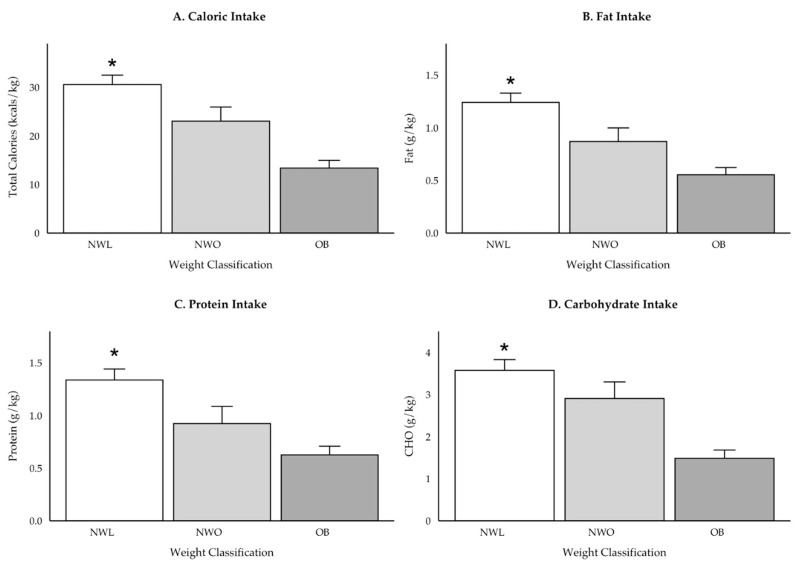
Relative caloric (**A**) and macronutrient (**B**) fat, (**C**) protein, (**D**) CHO intake by weight classification while accounting for sex. * denotes significant difference from OB (*p* < 0.001). Estimates and standard errors for NWL versus OB: Kcal·kg^−1^ = +17.1 (3.5), *p* < 0.001; Protein = +0.70 g·kg^−1^ (0.18) g·kg^−1^, *p* < 0.001; Fat = +0.73 (0.16) g·kg^−1^, *p* < 0.001; CHO = +2.0 (0.46) g·kg^−1^, *p* < 0.001.

**Figure 6 ijerph-22-00171-f006:**
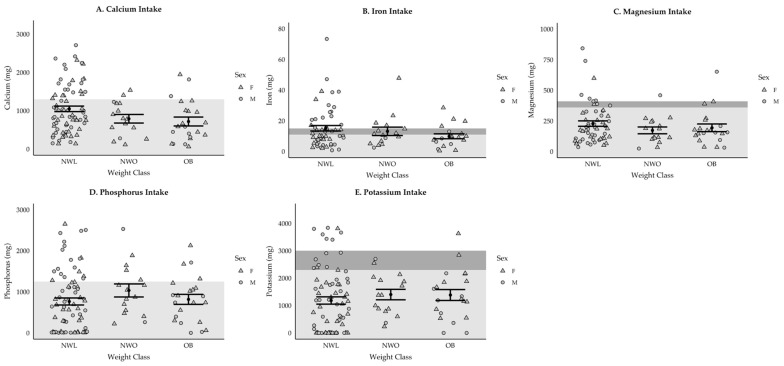
(**A**–**E**) Participants meeting the RDA for each mineral. Triangles represent girls and circles represent boys. Those who fall within the shaded area do not meet the RDA. The difference in RDA between boys and girls is shown by the dark gray shading. The iron RDA is higher for girls while the RDA for magnesium and potassium is higher for boys. For sodium, those in the shaded area are above the RDA.

**Table 1 ijerph-22-00171-t001:** Participant demographics.

		Total			Boys			Girls	
	NWL	NWO	OB	NWL	NWO	OB	NWL	NWO	OB
n	80	16	22	45	2	8	35	14	14
Prevalence (%)	67.8	13.6	18.6	81.8	1.8	14.5	55.6	22.2	22.2
Age (Yr)	16 ± 1	16 ± 1	16 ± 1	16 ± 1	17 ± 2	16 ± 1	16 ± 1	16 ± 1	16 ± 1
Height (cm)	173 ± 11	166 ± 7	168 ± 8	179 ± 10	170.5 ± 3.5	170 ± 11	166 ± 7	165 ± 7	168 ± 6
Weight (kg)	62.2 ± 10.7 *	63.4 ± 6.0 *	90.7 ± 14.7	66.3 ± 11.3 *	69.0 ± 0.8	88.3 ± 16.9	56.9 ± 7.2 *	62.6 ± 6.0 *	92.2 ± 13.7

Values are expressed as mean ± SD. NWL, normal weight lean; NWO, normal weight obese; OB, obese; Yr, year; cm, centimeter; kg, kilograms; * Indicates significant difference from OB (*p* < 0.05).

**Table 2 ijerph-22-00171-t002:** Relationship between macronutrients and body composition.

	All		Boys		Girls	
	BMI	BF (%)	BMI	BF (%)	BMI	BF (%)
Total Calories	−0.159	−0.279 ***	−0.038	−0.330 **	−0.309 *	−0.210
Calories per kg	−0.424 ***	−0.357 ***	−0.268 *	−0.430 ***	−0.607 ***	−0.458 ***
Protein (g)	−0.066	−0.302 ***	−0.140	−0.266 *	−0.266 *	−0.202
CHO (g)	−0.223 **	−0.237 **	−0.124	−0.290 *	−0.337 **	−0.194
Fat (g)	−0.107	−0.257 **	−0.070	−0.317 **	−0.262 *	−0.230
Protein (g·kg^−1^)	−0.332 ***	−0.384 ***	−0.150	−0.353 **	−0.527 ***	−0.435 ***
CHO (g·kg^−1^)	−0.454 ***	−0.304 ***	−0.336 **	−0.370 **	−0.610 ***	−0.426 ***
Fat (g·kg^−1^)	−0.339 ***	−0.324 ***	−0.196	−0.411 ***	−0.495 ***	−0.412 ***

Spearman’s correlations between macronutrients and BMI and BF percentage for the whole sample, boys, and girls. Significant relationships are denoted by * (*p* < 0.05). Significant relationships are denoted by * (*p* < 0.05), ** (*p* < 0.01), and *** (*p* < 0.001).

**Table 3 ijerph-22-00171-t003:** Absolute macronutrient intake.

		Total			Boys			Girls	
	NWL	NWO	OB	NWL	NWO	OB	NWL	NWO	OB
Total Calories	1872 ± 1068 *	1470 ± 783	1192 ± 667	2020 ± 1191	1232 ± 1191	1052 ± 600	1682 ± 864	1505 ± 767	1272 ± 712
Protein (g)	83.8 ± 62.2	59.3 ± 41.7	54.4 ± 31.4	95.4 ± 72.7	75.6 ± 87.0	56.0 ± 29.0	68.8 ± 41.7	57.0 ± 37.1	53.5 ± 33.7
Carbohydrate (g)	218 ± 137 *	187 ± 111	134 ± 84	238 ± 153	140 ± 150	107 ± 80	193 ± 112	193 ± 110	149 ± 85
Fat (g)	75.8 ± 47.8 *	55.4 ± 33.5	49.3 ± 28.8	78.3 ± 51.1	43.5 ± 31.0	44.7 ± 25.2	72.6 ± 43.5	57.1 ± 34.6	51.9 ± 31.3

Values are expressed as mean ± SD. NWL, normal weight lean; NWO, normal weight obese; OB, obese; g, grams. Significant differences (*p* < 0.05). * Indicates significant difference from OB.

**Table 4 ijerph-22-00171-t004:** Relationships between micronutrients and body composition.

	All		Boys		Girls	
	BMI	BF (%)	BMI	BF (%)	BMI	BF (%)
Sodium	−0.015	−0.103	−0.092	−0.311 **	−0.138	−0.140
Magnesium	−0.158	−0.207 *	0.055	−0.094	−0.386 **	−0.277 *
Calcium	−0.139	−0.269 **	−0.031	−0.207	−0.247 *	−0.139
Zinc	−0.123	−0.211 *	−0.067	−0.122	−0.314 **	−0.254 *
Iron	−0.144	−0.166	0.051	0.121	−0.363 **	−0.254 *

Spearman’s correlations between micronutrients and BMI and BF percentage for the whole sample, boys, and girls. Significant relationships are denoted by * (*p* < 0.05) and ** (*p* < 0.01).

## Data Availability

The data sets generated in the present study are available from the corresponding author upon request.

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
