# Peer review of "Adolescents with Normal Weight Obesity Have Less Dry Lean Mass Compared to Obese Counterparts"

_ijerph, 2025, doi:10.3390/ijerph22020171_

Round 1
Reviewer 1 Report
Comments and Suggestions for Authors
This study investigated the prevalence of normal weight obesity and dietary behaviors in adolescents (aged 14–19) from rural U.S. areas, Results revealed distinct dietary patterns and body composition differences between normal weight obesity, normal weight lean and obese groups, highlighting the need for further research on health implications in youth.
The study is interesting, well-conceived and provides interesting results that can be applied in a practical and future research sense.
- The time of conducting the study should be added to the abstract.
- The introduction is well written and clearly introduces the topic of the manuscript. I advise adding 1-2 sentences at the end that highlight why this paper is a novel.
- The methodology is written in sufficient detail. Perhaps I would advise separating the wholes into subheadings in order to have a better overview. Add the study period.
-In the results, replace P with lowercase p. For values ​​less than p<0.001, do not round to more decimal places. I recommend that the graphics be in color for better visibility. Correlations should be reported to 3 decimal places.
- The biggest complaint is that there is no research limitation indicated in the discussion. Write one full paragraph. The sample is relatively small, so it is necessary to highlight the strengths why they overcome the limitations.
- I would advise adding a few more references not older than 10 years.
Author Response
- The time of conducting the study should be added to the abstract.
Thank you, we have added the dates during which data were collected to the abstract.
- The introduction is well written and clearly introduces the topic of the manuscript. I advise adding 1-2 sentences at the end that highlight why this paper is a novel.
Thank you, we have added some information to the introduction to better set up our purpose (lines 79-81)
- The methodology is written in sufficient detail. Perhaps I would advise separating the wholes into subheadings in order to have a better overview. Add the study period.
Thank you, subheadings are now included as well as the dates during with data were collected.
-In the results, replace P with lowercase p. For values ​​less than p<0.001, do not round to more decimal places. I recommend that the graphics be in color for better visibility. Correlations should be reported to 3 decimal places.
Thank you, these changes to the p-values and correlations have been made. We are looking into submitting color graphics to the journal.
- The biggest complaint is that there is no research limitation indicated in the discussion. Write one full paragraph. The sample is relatively small, so it is necessary to highlight the strengths why they overcome the limitations.
Thank you, we have added a limitations section to highlight the disadvantages of our body composition method, the limitations of self-reported diet data, and our sample selection.
- I would advise adding a few more references not older than 10 years.
Conclusions
- Please remove the statement of the study’s purpose from the conclusion section.
Thank you, this is now removed.
Reviewer 2 Report
Comments and Suggestions for Authors
Dear authors;
First, I would like to commend you for conducting this important study on the prevalence of normal weight obesity and dietary behaviors among adolescents aged 14-19 years in the United States. This research provides valuable insights into a topic that is crucial for public health. However, there are several areas where further clarification or revision would strengthen the manuscript. In order for the paper to be considered for publication, I kindly ask that you address the specific comments outlined below.
Abstract
· Line 20: The term "DLM" for Dry Lean Mass is used only once in the abstract. It would be clearer to either eliminate the abbreviation or use the full term instead of "DLM" to avoid confusion.
· Line 25: Consider including a concrete, actionable recommendation based on the findings of the study to enhance the impact of the abstract.
Introduction
· The introduction provides a solid foundation for the study and effectively introduces the concept of normal weight obesity. However, I have a couple of minor suggestions to further improve clarity:
· Line 60: Please expand on the concept of poor diet quality, particularly its potential long-term effects and relevance to adolescent health.
· Line 72: It would be helpful to explicitly state the study hypothesis in this section to guide the reader’s understanding of the study’s objectives.
Materials and methods
· Subheadings: I strongly encourage the authors to organize this section with subheadings such as Participants, Study Design, Measurements, and Data Analysis. This would improve the clarity and flow of the methods section and help readers navigate through the different components of the study.
· Line 82: Please provide more detailed information about the ethical approval process, including the name of the ethics committee that approved the study.
· Line 116: It would be beneficial to explain how normality was assessed for the data.
Results
· The results section is generally well-presented. However, the inclusion of subheadings would further aid the reader in interpreting the findings. The tables and figures are concise and provide a clear overview of the results.
Discussion
· Overall, the discussion addresses the key findings of the study, but there are a few areas that need further attention:
· The first paragraph of the discussion largely reiterates information already presented in the introduction. I recommend removing this paragraph and directly starting with the interpretation of your main findings.
· Line 227: This paragraph requires further elaboration. It lacks sufficient depth in its current form. I suggest expanding on the implications of your findings and offering more critical analysis.
· Line 246: Similar to the previous comment, this paragraph also would benefit from a more critical interpretation. Please consider adding a deeper analysis and discussing potential limitations or alternative interpretations of your results. Additionally, I encourage you to structure the paragraphs more rigorously to improve the flow and clarity of the discussion.
Conclusions
· Please remove the statement of the study’s purpose from the conclusion section.
Comments on the Quality of English Language
N/A
Author Response
Abstract
- Line 20: The term "DLM" for Dry Lean Mass is used only once in the abstract. It would be clearer to either eliminate the abbreviation or use the full term instead of "DLM" to avoid confusion.
Thank you for pointing this out, we have removed the abbreviation from the abstract.
- Line 25: Consider including a concrete, actionable recommendation based on the findings of the study to enhance the impact of the abstract.
Thank you, we have added conclusions and a future direction.
Introduction
- The introduction provides a solid foundation for the study and effectively introduces the concept of normal weight obesity. However, I have a couple of minor suggestions to further improve clarity:
- Line 60: Please expand on the concept of poor diet quality, particularly its potential long-term effects and relevance to adolescent health.
Thank you, we have changed the wording for clarity to include “Low fruit and vegetable consumption coupled with high intake of refined, fried, and fatty foods” instead of “poor diet quality”.
- Line 72: It would be helpful to explicitly state the study hypothesis in this section to guide the reader’s understanding of the study’s objectives.
Thank you, the hypothesis has been added.
Materials and methods
- Subheadings: I strongly encourage the authors to organize this section with subheadings such as Participants, Study Design, Measurements, and Data Analysis. This would improve the clarity and flow of the methods section and help readers navigate through the different components of the study.
Thank you for this suggestion, we have now included subheadings.
- Line 82: Please provide more detailed information about the ethical approval process, including the name of the ethics committee that approved the study.
Thank you, we have added that the ethics committee was at the University of Idaho.
- Line 116: It would be beneficial to explain how normality was assessed for the data.
Thank you, we have added that normality was assessed using the Shapiro-Wilkes test and adjusted our correlations to reflect the non-normal distribution.
Results
- The results section is generally well-presented. However, the inclusion of subheadings would further aid the reader in interpreting the findings. The tables and figures are concise and provide a clear overview of the results.
Thank you, subheadings are included.
Discussion
- Overall, the discussion addresses the key findings of the study, but there are a few areas that need further attention:
- The first paragraph of the discussion largely reiterates information already presented in the introduction. I recommend removing this paragraph and directly starting with the interpretation of your main findings.
Thank you for this suggestion. We have removed the duplicate information.
- Line 227: This paragraph requires further elaboration. It lacks sufficient depth in its current form. I suggest expanding on the implications of your findings and offering more critical analysis.
Thank you, we have added some information to elaborate on our findings and the potential limitations in our dataset.
- Line 246: Similar to the previous comment, this paragraph also would benefit from a more critical interpretation. Please consider adding a deeper analysis and discussing potential limitations or alternative interpretations of your results. Additionally, I encourage you to structure the paragraphs more rigorously to improve the flow and clarity of the discussion.
Thank you, we have restructured some of the discussion and added a Strengths and Limitations section.